# Multi-agent System with Individual Optimized Expertise for Retrieval Augmented Generation

## Abstract

This paper studies the problem of retrieval-augmented generation (RAG), which leverages external knowledge to increase the performance of language models. Despite the remarkable progress, current RAG approaches are still far from satisfactory due to inadequate retrieval and potential hallucination. Towards this end, we propose a novel approach named Multi-agent System with Individual Optimized Expertise (MAPS) for RAG. The core idea of our MAPS is to equip three well-designed agents with different specializations using individual optimization corpus. In particular, we first expand ambiguous queries with a query extension agent, and quantify the reward based on the accuracy, which can be utilized to refine our agent for higher expansion efficiency. To enhance the retrieval quality, we include a unanimity voting to annotate the current query as insufficient or sufficient, and their generative outcomes are utilized as ground truth to supervised fine-tune the judge agent. To further mitigate potential hallucination, an answer agent is optimized with dynamic matching-based rewards with curriculum learning for final outputs. Extensive experiments across multiple benchmark datasets validate the effectiveness of the proposed MAPS in comparison with state-of-the-art approaches.

## 1 Introduction

Large language models (LLMs) have achieved remarkable success in natural language understanding and reasoning. By pre-training on a series of high-quality instruction datasets, LLMs can acquire a wide range of factual knowledge (Achiam et al., 2024). These pre-trained models demonstrate strong performance on many tasks, highlighting their capacity to understand and accurately respond to diverse queries (Xiao et al., 2025). Nevertheless, pre-trained models may fail to capture the most up-to-date information in highly specialized domains, particularly those involving private or sensitive knowledge, such as healthcare and law (Huang et al., 2025). This issue limits LLMs' capability in some knowledge-intensive tasks, even introducing severe hallucinations.

To enhance response quality, retrieval augmented generation (RAG) has been employed as a framework that leverages external knowledge to reduce reliance on the pre-trained LLMs' parametric knowledge (Gao et al., 2024). Typical RAG systems include two main steps: retrieval and generation. In the retrieval step, the key point is to locate the lacking knowledge and define a retrieval query to retrieve the knowledge from an external corpus. And then, in the generation stage, the retrieved contents are integrated with the original question and fed into the LLM. The augmented input provides the LLM with relevant knowledge and the latest information, thereby enhancing the quality of responses and reducing hallucinations (Mishra et al., 2024). The main advantage of RAG is that it provides external contextual information to answer

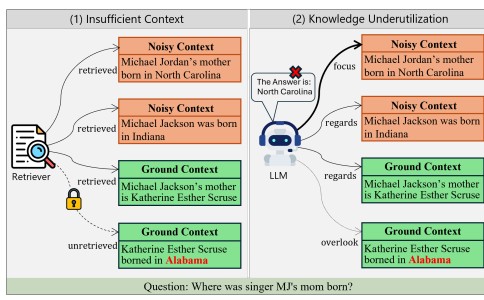

Figure 1: Problem illustration. RAG struggles with inefficient knowledge acquisition in the retrieval stage and knowledge underutilization in the answer stage.

questions when combined with the model's parametric knowledge (Yu et al., 2025). However, this advantage is constrained by two challenges. First, the system may not be able to accurately identify the missing knowledge and generate an effective query for retrieval. An inefficient query formulation may lead to the retrieval of numerous irrelevant or even biased documents, thereby increasing the model's reasoning burden (Yu et al., 2024a; Cuconasu et al., 2024). Second, even when relevant documents are retrieved, the model may be unable to appropriately utilize the provided context, which may lead to further hallucinations (Hsieh et al., 2024; Liu et al., 2024).

Existing approaches to optimizing the pipeline of RAG include the branching method (Kim et al., 2024; Shi et al., 2024), context rank (Yu et al., 2024b), and iterative retrieval (Asai et al., 2024; He et al., 2024). These approaches merely adjust the pipeline of RAG, remaining insufficient to counteract the noise or bias introduced by retrieved knowledge. Furthermore, many recent works fine-tune specific LLMs for the RAG pipeline to improve robustness and factual consistency in the generation stage. Such efforts include supervised fine-tuning with human-annotated datasets (Niu et al., 2024) and utilizing reinforcement learning to integrate reasoning ability with retrieval (Song et al., 2025). However, by placing the burden of generating retrieval queries and answering on the same backbone model, these approaches inherit the models' intrinsic biases, limiting their robustness and ability to generate effective queries. Recently, a growing trend is to break down complex processes into multiple specialized agents or modules that work collaboratively (Chen et al., 2024; Luo et al., 2025). Instead of a single model handling everything, each agent is responsible for a specific subtask in a multi-agent system (Bo et al., 2024). This paradigm has also been introduced into the RAG framework, resulting in a remarkable improvement in removing hallucinations and effectively utilizing external knowledge (Nguyen et al., 2025; Hu et al., 2025). However, existing multi-agent RAG frameworks still share the same backbone LLM without adaptive adjusting, which potentially undermines response reliability.

To address these challenges, we propose Multi-Agent System with Individually Optimized Expertise (MAPS), an innovative multi-agent collaborative RAG system that tightly couples the retrieval and generation stages to reduce hallucinations and improve retrieval quality. Unlike existing multi-agent RAG approaches that share a single backbone model (Hu et al., 2025), MAPS assigns task-adaptive LLMs to agents responsible for distinct subtasks, allowing each agent to specialize in its role. This targeted specialization enables the system to pinpoint missing knowledge and utilize retrieved knowledge more effectively, yielding higher answer quality. Because each agent operates on a narrow slice of the pipeline, MAPS also lowers training cost and improves reliability.

Our MAPS implements an adaptive multi-agent RAG framework that decomposes the workflow into three subtasks: retrieval query generation, sufficient information judgment, and answer generation, each subtask handled by an optimized expert agent. This adaptive design enables effective handling of diverse questions and yields robust performance across domains. In addition, we present a streamlined data-generation and training protocol for each agent that significantly improves effectiveness while maintaining modest supervision overhead. Crucially, the training data construction and fine-tuning procedures embed substantial cross-agent collaboration, which strengthens coordination within the multi-agent system, tightly couples the subtasks of the pipeline, and markedly enhances robustness. The protocol facilitates a straightforward transfer to a wide range of RAG scenarios and provides a practical path to enhance real-world RAG deployments. Our main contributions can be summarized as follows:

- *Problem Connection*. We propose MAPS, which assigns specialized agents to distinct stages of the RAG pipeline, thereby improving the efficiency of acquiring missing knowledge and significantly enhancing answer quality.

- *Novel Methodology*. Our MAPS introduces novel data generation methods that expand limited supervised data to align different subtasks of RAG, and further adopt adaptive fine-tuning paradigms for each agent. This design helps enhance retrieval efficiency and suppress hallucinations.

- *High Performance*. Comprehensive experiments across the latest benchmarks demonstrate that our MAPS outperforms a range of competitive baselines. In addition, the framework can be readily extended to various RAG scenarios.

## 2 RELATED WORK

**Retrieval-augmented Generation.** Large language models have achieved strong performance across diverse tasks. However, they remain susceptible to hallucinations and insufficient context.

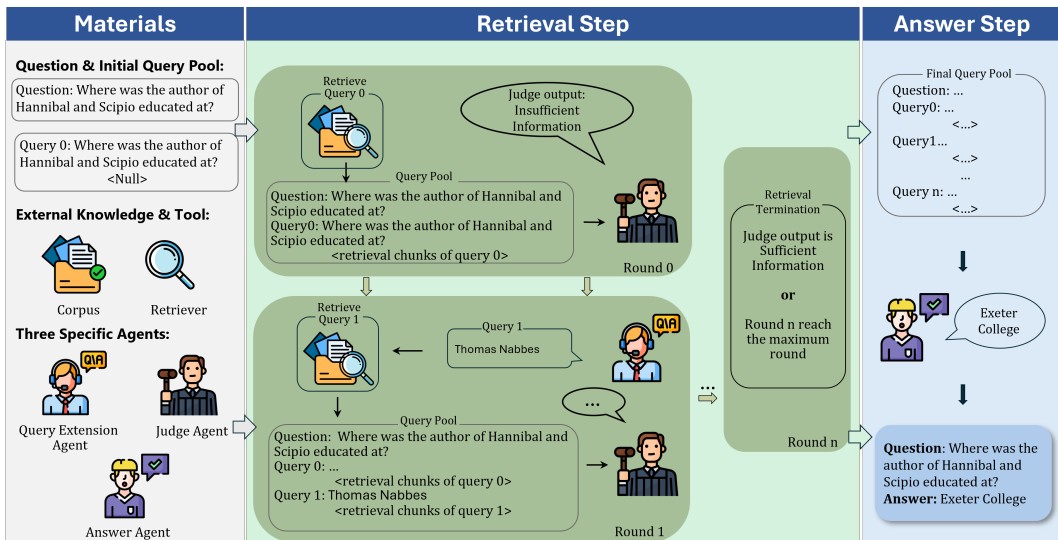

Figure 2: An overview of our MAPS framework. The cooperation of three specific agents improves the retrieval and answer quality.

Retrieval augmented generation addresses this limitation by retrieving external evidence and conditioning the model's output on the retrieved context (Fan et al., 2024). Early RAG pipelines use the original input as a query to search on an external corpus, and then integrate the retrieval context into the model's input as supplementary knowledge (Gao et al., 2024). To meet more complex retrieval needs, many methods introduce iterative retrieval generation branching (Kim et al., 2024) or loops (Yoran et al., 2024; Chan et al., 2024; Jiang et al., 2023; Shao et al., 2023), letting the model adapt its retrieval strategy when evidence is insufficient. With the rise of chain-of-thought prompting (Zhang et al., 2025b), reasoning traces have also been used to improve retrieval efficiency and recall (Li et al., 2025; Trivedi et al., 2023). Some multi-agent methods have also been explored in RAG to enhance retrieval quality (Hu et al., 2025; Nguyen et al., 2025). Despite these advances, most existing methods still share a single backbone model across all roles in the RAG pipeline and lack targeted fine-tuning, which limits further performance gains.

**Multi-agent System.** Recent work shows that multi-agent interaction can substantially enhance LLM capabilities across three main regimes (Agashe et al., 2025). First, debate-based frameworks run adversarial dialogues or critiques in which agents challenge each other to refine reasoning and factuality (Du et al., 2024; Zhang et al., 2025a). Second, ensemble coordination runs multiple agents in parallel with minimal communication and aggregates their outputs via voting, validation, or game-theoretic consensus (Wang et al., 2025; Yue et al., 2024). Third, role-specialized collaboration assigns complementary roles to decompose problems and iteratively refine solutions (Wei et al., 2025; Liu et al., 2025). However, the multi-agent setting for RAG remains underexplored. Contemporary RAG pipelines typically rely on a single LLM to both retrieve and generate, with limited support for collaboration among specialized agents.

## 3 THE PROPOSED MAPS

**Problem Definition.** Retrieval augmented generation enhances large language models by integrating external knowledge as context that is not stored in the model's parameters. Given a collection of retrieval text chunks $C$ and a corresponding retrieval query $Q$, which was identified by the LLM to obtain knowledge from an external corpus. If necessary, it can perform multiple retrieval rounds to acquire additional knowledge. However, excessive context may introduce noise and increase the risk of hallucination. Our objective is to build an efficient multi-agent RAG system, where each agent is specialized and fine-tuned to a specific subtask within the RAG pipeline. The multi-agent RAG system should formulate a new retrieval query that targets the missing evidence in an external corpus to help answer question $x$ while minimizing irrelevant results. The answer agent must coordinate with the other agents, utilizing the acquired context to generate an accurate response. The central problem is how to train multiple agents to cooperate across retrieval, judgment, and answer generation subtasks, so that they function as an effective whole.

### 3.1 FRAMEWORK OVERVIEW

We address the limitation of existing RAG systems, which rely on a single pretrained LLM across all stages, leading to weak coordination between components and making it hard to excel at multiple subtasks. In MAPS, three specialized agents are trained to adapt to their respective subtasks while engaging in substantial cross-agent cooperation. This training strategy strengthens coordination within the multi-agent RAG system, tightly couples the subtasks, and improves robustness and factual consistency. In particular, during the construction of training data and the training process, the query agent is trained to retrieve evidence aligned with the preferences of the answer agent, making it easier to produce accurate and unbiased answers. The judge agent is trained to understand the other agents' behaviors and capabilities, enabling more reliable decisions about whether the current retrieval evidence is sufficient.

### 3.2 ANSWER AGENT OPTIMIZATION WITH MATCHING-BASED REINFORCEMENT LEARNING

The answer agent is a key component responsible for using the retrieved documents to answer questions with high accuracy. The primary goal for training the answer agent is to enhance the utilization of retrieval content and avoid hallucinations. To train the agent, we build a training corpus that spans different numbers of retrieval steps and various chunk set sizes, enabling the agent to learn to ignore biases in the context and focus on the key evidence.

Specifically, we first sample training questions for the answer agent from the multi-hop QA datasets HotpotQA (Yang et al., 2018) and 2WikiMultiHopQA (Ho et al., 2020). And then we assign difficulty labels (easy/medium/hard) and retain 8148 questions from the medium and hard splits following (Song et al., 2025). We then build a retrieval-only debate RAG pipeline (Hu et al., 2025) with Llama-3-8B-Instruct, and cap the number of retrieval rounds at three. For the first 6000 questions, we retrieve the top 3 chunks per round; for the remaining questions, we use the top 5. The debate RAG method has three agents to iteratively refine retrieval and retrieved contents. We store the query pool and the retrieved chunks at the end of each debate round and treat each snapshot as a training sample. For each question, we create 1–4 samples by varying the number of collected queries and chunks. This setup pushes the agent to ignore noise and rely on the key evidence under different context sizes. The details of the answer training data generation and debate RAG prompts are shown in Appendix B.

To improve the generative capabilities of the answer agent, we propose a two-stage outcome-based reinforcement learning (RL) training method based on the GRPO algorithm (Shao et al., 2024). In Stage 1, the model is trained to utilize the provided information and minimize internal hallucinations, using questions with top 3 retrieval chunks for each retrieval query. Specifically, an outcome-based reward consists of an exact match (EM) score and a precision score, which are used in this stage. The EM is defined as follows:

$$EM(y, y^*) = \begin{cases} 1, & \text{if the } y \text{ exact matchs the } y^* \\ 0, & \text{else,} \end{cases} \quad (1)$$

where $y$ is the predicted answer generated by the answer agent and $y*$ is the gold answer of the input question $x$. The precision score is:

$$PR(y, y^*) = \frac{IN(y, y^*)}{PN(y)}, \quad (2)$$

where the $PN(y)$ represents the word count of the answer, $IN$ indicates the word count of the intersection between the two answers, and $\alpha$ is a tuned hyper-parameter. To further discourage explanations and other extraneous content in the answer, we add a format term to the reward, defined as:

$$L(y) = \begin{cases} -1, & \text{if the length of } y \text{ larger than } l \\ 0, & \text{else,} \end{cases} \quad (3)$$

where $l$ is a tuned hyper-parameter. Therefore, the final reward $R_{a1}$ of stage 1 is the sum of the outcome metrics and format metrics:

$$R_{a1} = EM(y, y^*) + \alpha * PR(y, y^*) + L(y), \quad (4)$$

where $\alpha$ is a tuned hyper-parameter.

In stage 2, the answer agent is trained to effectively locate key evidence in richer contexts and to explore broader reasoning paths, using questions with the top 5 retrieval chunks for each retrieval query. The outcome-based reward in this stage comprises an exact match(EM) score and an F1 score, which is defined as follows:

$$R_{a2} = EM(y, y^*) + \alpha * F1(yy^*) + L(y),$$ (5)

where the $F1$ score is:

$$F1(y, y^*) = \frac{2 * IN(y, y^*)}{PN(y) + PN(y^*)}.$$ (6)

---

**Prompt for Answer Agent in Training**

```
<|eot_id|><|start_header_id|>system<|end_header_id|>
```

Answer the question based on the given document. Output only the final answer with no explanations or additional text.

EXISTING QUERIES and RETRIEVED DOCUMENTS

Query 1: Are the directors of both films I Can Do Bad All By Myself (Film) and Shit Year from the same country?

Retrieved Content:

Doc 1(Title: "I Can Do Bad All by Myself (film) "): ...

Doc 2(Title: "I Can Do Bad All by Myself (film) ") : ...

Doc 3(Title: "Shit Year") : ...

Doc 4(Title: "Shit Year"): ...

Doc 5(Title: Themselves): ...

Query 2: Tyler Perry

Retrieved Content:

Doc 1(Title: "Tyler Perry"): ...

Doc 1(Title: "Tyler Perry"): ...

Doc 2(Title: "Tyler Perry Studios"): ...

Doc 3(Title: "Tyler Perry"): ...

Doc 4(Title: "Tyler Perry Studios"): ...

Doc 5(Title: "Tyler Perry"): ...

```
<|eot_id|><|start_header_id|>user<|end_header_id|>
```

Question: Are the directors of both films I Can Do Bad All By Myself (Film) and Shit Year from the same country?

```
<|eot_id|><|start_header_id|>assistant<|end_header_id|>
```

---

### 3.3 QUERY EXTENSION AGENT OPTIMIZATION WITH OUTCOME-BASED REWARD

The goal of retrieval-query generation in RAG is to precisely identify the gap in the current context and generate a new retrieval query that reduces whole context bias and thus obtains key evidence needed to answer the question. Existing methods struggle with a lack of a clear signal to judge the quality of a single query. In MAPS, we train an adaptive query extension agent to improve retrieval quality based on the answer agent's generating quality.

For training data generation, we sample questions from HotpotQA and 2WikiMultiHopQA, using each question as the retrieval query to get the corresponding training sample, obtaining the query refinement dataset as the training dataset. After obtaining the strong answer agent in section 3.2, we define the quality of an individual query by the change in answer quality after adding the chunk and before. As with the answer agent, we train the query-extension agent using GRPO, with the following reward:

$$R_q = F1(y_{n+1}, y^*) - F1(y_n, y^*),$$ (7)

where $y_n$ is the answer generated by the answer agent given the chunks retrieved in the first $n$ rounds. The training method encourages the query extension agent to pinpoint missing evidence under varied contexts and to generate effective queries accordingly.

### 3.4 Judge Agent Optimization with Inductive Supervised Fine-tuning

To decide whether the query extension agent has retrieved sufficient evidence and when to stop retrieval and begin answering, thereby limiting hallucinations from long contexts. We introduce a data-generation scheme and train the judge agent with it. The judge agent evaluates whether the retrieved context is sufficient to answer the question.

As with the query extension agent, we use the answer agent's response quality as the sole supervision signal for judging context quality. Unlike the query extension agent, the judge is trained via supervised fine-tuning (Joren et al., 2025). For training data generation, we also sample questions from HotpotQA and 2WikiMultiHopQA and use the same Debate RAG method with Section 3.2 to generate judge training samples. In order to get the supervised signal for the judge agent, we follow the test-time labeling method in (Zuo et al., 2025), run the answer agent on each sample, and draw 16 stochastic predicted answers. If all 16 are correct, we label the sample as **'Sufficient Information'**; if all 16 are incorrect, we label it **'Insufficient Information'**. This procedure yields reliable supervised signals, helping the judge learn to assess context sufficiency across various domains and context lengths. Finally, we combine these labeled samples to form the judge enhancement dataset, which is used to train the judge agent via supervised fine-tuning (SFT).

### 3.5 Summarization

With well-designed training methods, each agent within MAPS coordinates closely to generate high-quality answers. The optimization of MAPS is summarized in Algorithm 1. At inference time, the three agents collaborate to acquire external knowledge and generate the final answer. The pipeline has two main steps: the retrieval step and the answer step. The standard RAG methods often suffer from insufficient or irrelevant retrieval chunks in the retrieval process. To address this, MAPS employs a query extension agent and a judge agent to cooperate in the retrieval process. The multi-agent workflow begins by putting the raw question to the retriever to obtain the initial chunks from the external corpus:

$$Q_0, C_0 = \mathcal{R}(x, k), \tag{8}$$

where $Q_0$ and $C_0$ represent the initial query pool and its corresponding set of retrieved chunks for question $x$. The retriever $\mathcal{R}$ maps a query pool $Q$ to the union of the top-k chunks returned from the corpus for each query. In the iterative retrieval steps, each round begins with the *query extension agent* $\mathcal{A}_q$ using the previous query pool and retrieval chunks $(Q_n, C_n)$ to identify the knowledge gaps in answering the question $x$ and generate a new query. And then send the query to the retriever, to a new query pool $Q_{n+1}$ and corresponding retrieved chunks:

$$Q_{n+1}, C_{n+1} = Q_n + \mathcal{R}(\mathcal{A}_q(x, Q_n, C_n)). \tag{9}$$

At the end of each retrieval round, the *judge agent* $\mathcal{A}_j$ judges whether the current query pool is sufficient to answer the question $x$:

$$D = \mathcal{A}_j(x, Q_{n+1}, C_{n+1}). \tag{10}$$

If the decision $D$ is that the current information is insufficient to answer the question, we iterate to the next round(up to a fixed round budget). Otherwise, we switch to the answer step in which the *answer agent* generates the final answer based on the optimized query pool and their corresponding retrieved chunks:

$$y = \mathcal{A}_g(x, Q^*, C^*), \tag{11}$$

where $\mathcal{A}_g$ denotes the answer agent, and y is the final answer to the question $x$. All the prompts for the three agents are shown in Appendix A.

## 4 Experiment

In this section, we present the overall experimental workflow, including the experimental setup (Section 4.1) and empirical results and analysis (Section 4.2). To validate the effectiveness of MAPS,

---

**Algorithm 1:** Training Algorithm of MAPS

---

**Require:** Question dataset $Q$, Retriever $\mathcal{R}$, Corpus $C$ and Backbone model $L$

**Ensure:** Query-Extension agent $\mathcal{A}_q$, Judge agent $\mathcal{A}_j$ and Answer agent $\mathcal{A}_g$

  1. Extract $Q_1$ (hard questions) and $Q_2$ (normal questions) from $Q$;

  2. Conduct the retrieval debate method with $\mathcal{R}$ and $C$ on $Q_1$, build training dataset $T_1$;

  3. Train $L$ on $T_1$ with GRPO to obtain $\mathcal{A}_g$, using reward functions in Eq. 4 and Eq. 5;

  4. Conduct a single retrieval using question on $Q_2$ with $\mathcal{R}$ and $C$, build training dataset $T_2$;

  5. Train $L$ on $T_2$ with GRPO to obtain $\mathcal{A}_q$, using the reward in Eq. 7;

  6. Use $\mathcal{A}_g$ to answer all questions in $T_2$ with 16 samples, constructing training dataset $T_3$;

  7. Supervised fine-tuning of $L$ on $T_3$, obtaining the judge agent $\mathcal{A}_j$.

**Stop**

---

we design a comprehensive experiment that includes both in-domain and out-of-domain scenarios. We further assess the contribution of each specific agent through an ablation study. Additionally, we discuss the various settings in the inference and training of MAPS.

## 4.1 EXPERIMENTAL SETTINGS

**Baselines**. We implement native generation and standard RAG (Jin et al., 2025b) which represents the basic RAG method without optimization, as basic baselines. Moreover, we compare MAPS with several recent RAG frameworks, including branching method SuRE (Kim et al., 2024), loop method Self-RAG (Asai et al., 2024), IRCoT (Trivedi et al., 2023), and Iter-Retgen (Shao et al., 2023), R1 style reasoning RAG method Search-R1 (Jin et al., 2025a), R1-Searcher (Song et al., 2025), and multi-agent debate method DRAG (Hu et al., 2025).

**Datasets & Evaluation metrics**. Experiments are conducted on the dev datasets of four benchmarks: HotpotQA (Yang et al., 2018), 2WikiMultiHopQA (Ho et al., 2020), Musique (Trivedi et al., 2022), and Bamboogle (Press et al., 2023). HotpotQA and 2WikiMultiHopQA are in-domain datasets because parts of their training sets are used to train our agents and other training-based baselines. Musique and Bamboogle are out-of-domain datasets that are only used for evaluation. For evaluation metrics, following state-of-the-art RAG works (Hu et al., 2025; Chang et al., 2025; Jin et al., 2025a), we report exact match score and token-level F1 score metrics in the results.

**Implementation Details**. All compared models are reproduced and evaluated using FlashRAG (Jin et al., 2025b), and baseline backbones are used as specified in their original papers. We employed E5-base-v2 (Wang et al., 2024) as the retriever, and the retrieval corpus comprises the English Wikipedia as provided by the Wikipedia dump 2018 (Karpukhin et al., 2020), segmented into 100-word passages with appended titles. For each question, the external evidence is capped at 20 retrieved chunks. In the training processes of our MAPS, the backbone model is Llama-3.1-8B-Instruct. We select questions from the training sets of HotpotQA and 2WikiMultiHopQA to generate training samples (see Section 3). The answer agent is trained on 18380 samples generated from 8148 questions, using the GRPO algorithm. The query extension agent is trained on 10000 samples with GRPO. Each sample is rolled out 16 times under GRPO. Train batch size is 512, rollout batch size is 32, with the learning rate set to $10^{-6}$ and the KL divergence set to $10^{-4}$. The judge agent is trained on 44473 samples using supervised fine-tuning. Training is conducted on a single node with two H100 GPUs. For the query-extension agent, reward computation is offloaded to two additional L40 GPUs. All trained models for three agents are available at https://huggingface.co/Angus998/MAPS. More details are provided in Appendix A.

## 4.2 EMPIRICAL RESULTS AND ANALYSIS

**Comparison with Baselines**. We conduct comprehensive experiments to evaluate the performance of MAPS against various baselines across four benchmark datasets, reporting the EM and F1 scores achieved by our MAPS and other state-of-the-art methods. As reported in Table 1, MAPS achieves strong and consistent improvements on all benchmarks, demonstrating its robustness in settings that demand complex retrieval and multi-step inference. MAPS even surpasses recent R1-style reasoning-based RAG methods. On average, MAPS improves EM by 5.81% and F1 by 9.92% over

Table 1: The overall evaluation results of MAPS and other baselines on four benchmarks. **Bold** marks the best-performing method, and underline represents the second-best-performing method. All methods are evaluated under the same settings. Our MAPS achieves the strongest performance.

| Method | In Domain | | | | Out of Domain | | | |
| --- | --- | --- | --- | --- | --- | --- | --- | --- |
| | HotpotQA | | 2Wiki | | MuSiQue | | Bamboogle | |
| | EM | F1 | EM | F1 | EM | F1 | EM | F1 |
| *Basic RAG Method* | | | | | | | | |
| Native Gen | 16.19 | 23.80 | 8.23 | 16.25 | 1.61 | 6.09 | 16.80 | 24.18 |
| Standard RAG | 30.95 | 41.31 | 16.39 | 25.84 | 5.33 | 10.63 | 16.00 | 25.02 |
| *Without Training* | | | | | | | | |
| SuRe | 22.56 | 35.38 | 11.66 | 19.17 | 6.53 | 12.96 | 16.80 | 27.27 |
| IRCOT | 18.41 | 28.25 | 12.66 | 23.63 | 6.90 | 12.44 | 20.80 | 31.12 |
| Iter-RerGen | 33.50 | 44.19 | 16.51 | 26.36 | 7.24 | 12.97 | 19.20 | 27.01 |
| DRAG | 30.60 | 41.46 | 20.94 | 27.91 | 11.71 | 20.10 | 28.80 | 40.30 |
| *With Training* | | | | | | | | |
| Self-RAG | 15.96 | 28.42 | 11.59 | 23.51 | 4.22 | 11.95 | 6.40 | 15.76 |
| Search-R1 | 41.16 | 53.06 | 43.01 | 48.79 | 18.36 | 26.08 | 46.40 | 57.80 |
| R1-Searcher | 40.12 | 52.00 | 40.46 | 44.81 | 17.79 | 26.17 | 46.40 | 57.82 |
| MAPS | **43.50** | **57.70** | **47.10** | **56.60** | **18.57** | **28.18** | **49.60** | **62.00** |

the second-best method. This superior performance can be attributed to MAPS's decomposition of the RAG pipeline into smaller sub-tasks and performing each with a well-designed agent, which makes both the retrieval and answering more effective. In particular, compared to the other multi-agent baseline DRAG, MAPS improves EM by 74.47% and F1 by 59.00%, indicating that targeted fine-tuning for each agent design substantially boosts multi-agent RAG. In addition, we observe that MAPS achieves strong results on the musique and bamboogle datasets without any training on these benchmarks. This suggests that the agents learn effective retrieval and coordination under our diverse training scheme, yielding robust performance on new test sets.

Table 2: Ablation studies on the four benchmarks. All variants perform well below our MAPS.

| Variants | In Domain | | | | Out of Domain | | | |
| --- | --- | --- | --- | --- | --- | --- | --- | --- |
| | HotpotQA | | 2Wiki | | Musique | | Bamboogle | |
| | EM | F1 | EM | F1 | EM | F1 | EM | F1 |
| MAPS w/o query agent | 42.35 (↓2.72%) | 55.55 (↓5.08%) | 45.47 (↓3.85%) | 51.91 (↓11.07%) | 13.57 (↓9.37%) | 24.56 (↓8.55%) | 45.60 (↓9.45%) | 55.90 (↓14.40%) |
| MAPS w/o judge agent | 42.84 (↓1.54%) | 55.87 (↓3.28%) | 47.65 (↑1.15%) | 54.16 (↓4.51%) | 15.93 (↓10.11%) | 26.35 (↓6.94%) | 44.80 (↓10.71%) | 56.43 (↓9.87%) |
| MAPS w/o answer agent | 37.35 (↓16.47%) | 49.26 (↓17.13%) | 27.78 (↓69.55%) | 36.18 (↓56.44%) | 10.51 (↓66.89%) | 18.55 (↓51.91%) | 28.00 (↓77.14%) | 39.67 (↓56.29%) |
| MAPS | 43.50 | 57.70 | 47.10 | 56.60 | 18.57 | 28.18 | 49.60 | 62.00 |

**Ablation study.** In order to evaluate the impact of each specific agent in MAPS, we conduct ablation studies on the four benchmarks. Specifically, we evaluate three variants: (1) MAPS w/o query extension agent: replace the trained query extension agent with the base LLM. (2) MAPS w/o judge agent: replace the trained judge with the base LLM. (3) MAPS w/o answer agent: replace the trained answer agent with the base LLM. The experimental results are reported in Table 2. On average, all variants perform well below MAPS, with a larger drop on out-of-domain datasets than on in-domain datasets. This indicates that each specialized agent is effective and that collaboration among the three jointly trained agents further improves performance, especially on unseen data. Specifically, MAPS w/o answer agent performs worst on all benchmarks, indicating that a well-trained answer agent is critical for multi-agent RAG systems. MAPS w/o judge agent is the strongest among the variants and even shows a slight EM gain over MAPS on 2WikiMultiHopQA (+1.15%). But MAPS w/o judge agent still present worse performance on out-of-domain datasets than in-domain datasets, suggesting that the SFT training method for the judge agent potentially leads to overfitting. Moreover, although all variants underperform MAPS, they still exceed the basic RAG method and even some recent RAG methods, reinforcing that specialized multi-agent designs strengthen RAG. These results suggest that targeted training enables effective coordination, thereby reducing hallucinations and improving factual accuracy.

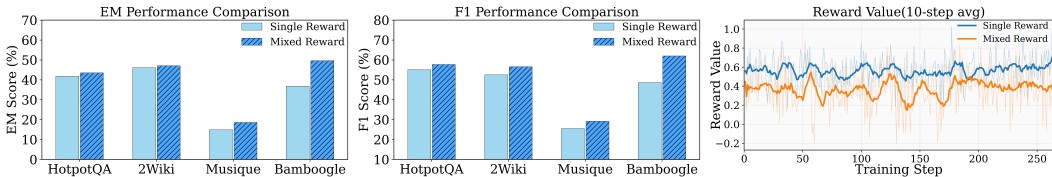

Figure 3: Visualization of EM and F1 metrics for MAPS with single or mixed reward training in the left two columns. The reward values calculated using a 10-step average of the last 264 steps in the training process are shown in the third column. Mixed reward design surpasses single reward design in overall performance.

**Reward design analysis.** To investigate the impact of reward design on MAPS, we train the answer agent with a single reward (see Equation 5) and the mixed reward design from Section 3.2, using the same Llama-3.1-8B-Instruct as backbone model. The results and reward values in the last 264 steps are shown in Figure 3. We find that single reward design yields a small drop(-5.9% on average) in two in-domain datasets and a much larger drop in two out-of-domain datasets (-17.03% on average), indicating that the two-stage Mix Reward improves overall performance, especially generalization. Comparing reward values over the last 264 training steps, where the reward functions are the same for both designs, further supports this finding. The single reward method achieves higher training rewards than the mixed reward design. This pattern suggests that the training model with a single reward is prone to overfitting the training distribution. We conclude that using a mixed reward design to train the agent reduces overfitting and improves generalization to unseen data.

**Analysis of retrieval rounds.** To systematically assess the impact of different retrieval rounds on the performance of MAPS. We fix the number of rounds in the retrieval step from 1 to 5 and evaluate them on the Musique and Bamboogle datasets. The results are shown in Table 3, where the flexible round is MAPS with a maximum round equal to 3, and the end decision is made by the judge agent. Fixing the round count reduces performance across all variants. On the musique dataset, as the re-

Table 3: MAPS performance with different iteration rounds on the retrieval step.

| Method | Musique | | Bamboogle | |
|---|---|---|---|---|
| | EM | F1 | EM | F1 |
| Round = 1 | 15.56 | 25.97 | 44.80 | 56.14 |
| Round = 2 | 15.85 | 26.09 | 44.80 | 55.87 |
| Round = 3 | 15.85 | 25.99 | 45.60 | 58.11 |
| Round = 4 | 15.47 | 25.61 | 45.60 | 58.16 |
| Round = 5 | 15.31 | 25.16 | 45.60 | 58.18 |
| Flexible Round | **18.57** | **28.18** | **49.60** | **62.00** |

trieval rounds increase, F1 increases slightly and then declines. On the bamboogle dataset, F1 shows a large jump when round is equal to 3 and only minor gains thereafter. These trends suggest that longer contexts can help but also introduce noise, increasing the risk of hallucination. The flexible setting achieves the best results, indicating that supplying an appropriate number of retrieval chunks is beneficial and that the judge agent effectively decides when to stop.

## 5 CONCLUSION

In this paper, we present MAPS, a framework that improves RAG by coordinating three individually optimized agents. MAPS decomposes the RAG pipeline into three sub-tasks: query extension, judging the sufficiency of the retrieved context, and generating the answer. We firstly design data generation methods for each subtask, all starting from question–answer supervision only, and construct three agent-specific training corpora with distinct signals that respond to different subtasks. And then we optimize each agent using reinforcement learning or supervised fine-tuning with its specific corpus. The experiment results show that our MAPS achieves superior performance among all state-of-the-art baselines on all benchmarks. Further analysis reveals that our constructed subtask-specific training corpora and mixed reward functions enhance each agent's ability to perform its role, and effective coordination among agents is crucial for reducing hallucinations and improving answer quality. This work enhances RAG answer quality and can be easily extended to various RAG scenarios, benefiting real-world LLM engineers. Moreover, our work provides a new direction for incorporating the cooperation of multiple specific agents into the RAG system.

One limitation of our work is that MAPS may need larger graphics memory to load the three agents. We would explore reducing the training and inference cost with lighter methods such as LoRA-based fine-tuning. In addition, exploring different dramatic reward designs may further enhance the robustness and generalization, making MAPS suitable for more complex tasks.

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

# A   PROMPTS AND DETAILS OF MAPS

## A.1   PROMPTS OF INFERENCE

---

**Prompt for Query Extension Agent**

**System**
You are a retrieval query generation agent.
Input:
QUESTION: ...
EXISTING QUERIES and RETRIEVED DOCUMENTS: ...

Task:
Generate exactly one NEW QUERY that:
1) Directly helps answer the QUESTION or fill gaps not covered by EXISTING QUERIES and RE-TRIEVED DOCUMENTS;
2) Is different from all EXISTING QUERIES.

Output rules:
Only keywords; no sentences, stopwords, or connectors (e.g., and, or, of, the, about).
Separate keywords with a single space;
Maximum 8 keywords.
Output: Only the NEW QUERY text. No explanations or extra text.

**User**
   Question: —*Input Question*—
   Query Pool: —*Current Query Pool*—

---

**Prompt for Judge Agent**

**System**
Determine whether the EXISTING QUERIES and RETRIEVED DOCUMENTS provide sufficient information to answer the QUESTION correctly. Output only one of the following: 'Sufficient Information' or 'Insufficient Information'. Do not include explanations or additional text.

**User**
   Question: —*Input Question*—
   Query Pool: —*Current Query Pool*—

---

**Prompt for Answer Agent**

**System**
Answer the question based on the given document. Output only the final answer with no explanations or additional text.
Query Pool:
—*Current Query Pool*—

**User**
   Question: —*Input Question*—

---

## A.2 PROMPTS IN TRAINING

| Prompt for Query Extension Agent in Training |
| --- |
| `<\|begin_of_text\|><\|start_header_id\|>system<\|end_header_id\|>`
You are a retrieval query generation agent.
Input:
QUESTION: ...
EXISTING QUERIES and RETRIEVED DOCUMENTS: ...

Task:
Generate exactly one NEW QUERY that:
1) Directly helps answer the QUESTION or fill gaps not covered by EXISTING QUERIES and RE-
TRIEVED DOCUMENTS;
2) Is different from all EXISTING QUERIES.

Output rules:
Only keywords; no sentences, stopwords, or connectors (e.g., and, or, of, the, about).
Separate keywords with a single space;
Maximum 8 keywords.
Output: Only the NEW QUERY text. No explanations or extra text.
`<\|eot_id\|><\|start_header_id\|>user<\|end_header_id\|>`

Question: Are the directors of both films I Can Do Bad All By Myself (Film) and Shit Year from the same
country?
EXISTING QUERIES and RETRIEVED DOCUMENTS
Query 1: Are the directors of both films I Can Do Bad All By Myself (Film) and Shit Year from the same
country?
Retrieved Content:
Doc 1: ...
Doc 2: ...
Doc 3: ...
Doc 4: ...
Doc 5: ...
`<\|eot_id\|><\|start_header_id\|>assistant<\|end_header_id\|>` |

---

**Prompt for Judge Agent in Training**

```
<|eot_id|><|start_header_id|>system<|end_header_id|>
```

**System**

Determine whether the EXISTING QUERIES and RETRIEVED DOCUMENTS provide sufficient information to answer the QUESTION correctly. Output only one of the following: 'Sufficient Information' or 'Insufficient Information'. Do not include explanations or additional text.

```
<|eot_id|><|start_header_id|>user<|end_header_id|>
```

Question: Are the directors of both films I Can Do Bad All By Myself (Film) and Shit Year from the same country?

EXISTING QUERIES and RETRIEVED DOCUMENTS

Query 1: Are the directors of both films I Can Do Bad All By Myself (Film) and Shit Year from the same country?

Retrieved Content:

Doc 1: ...

Doc 2: ...

Doc 3: ...

Doc 4: ...

Doc 5: ...

```
<|eot_id|><|start_header_id|>assistant<|end_header_id|>
```

'Insufficient Information'

---

## B PROMPTS OF TRAINING DATA GENERATION

---

**Prompt for Proponent in Retrieval Debate**

**System**

You are a debater. Argue that the current retrieved content is sufficient to answer the question and try to give the answer based on the given documents. Deliver a brief, strong argument with clear reasoning. Do not suggest further retrieval. No extra explanations.

**User**

Question: —*Input Question*—
Query Pool: —*Current Query Pool*—

---

**Prompt for Opponent in Retrieval Debate**

**System**

You are a critical thinker and debater. Your task is to challenge the sufficiency of the current documents. However, your ultimate goal is to find the correct answer efficiently. If you believe the provided information is TRULY and COMPLETELY sufficient, your duty is to concede.

The action you can choose:

1. Query Expansion: If a completely new line of inquiry is needed. Output exactly in this format in the end of your response: Query Expansion: [New Query]

2. Concede: If the current information is truly sufficient. Only Output: Concede

Deliver a brief, strong argument, then you must choose one action in the exact format required.

**User**

Question: —*Input Question*—
Query Pool: —*Current Query Pool*—

---

**Prompt for Moderator in Retrieval Debate**

**System**

You are the judge in a debate. Your task is to evaluate the arguments from agents. There are two types of agents:

1. Proponent Agent: Argues that the current retrieved content is sufficient.

2. Opponent Agent: Argues that the current retrieved content is insufficient and proposes query refinement.

Question: —*Input Question*—
Query Pool: —*Current Query Pool*—
Proponent:
—*Output of Proponent*—
Opponent:
—*Output of Opponent*—

Output only the agent's name. Do NOT output more than one agent or any explanation.

---

# C   CASE STUDY

Table 4: Case study illustrating a step-by-step imperfect answer generating process of MAPS

| **Case Study 1** |
|---|
| **Question:** Who played the character in the Santa Clause 3 that has a series named after it that includes Frost at Christmas? |
| **Ground truth:** Martin Short |
| **Retrieval Round 0:** |
| (Using question as query to retrieve) |
| **Query Pool:**{ |
|     Query 0: Who played the character in the Santa Clause 3 that has a series named after it that includes Frost at Christmas? |
|     Retrieved Chunks for Query 0: |
|       (1) ID: 9979537;     Title:The Santa Clause 3 ;     Content: ... |
|       (2) ID: 9738449;     Title:The Santa Clause (film series) ;     Content: ... |
|       (3) ID: 9979546;     Title:The Santa Clause 3: The Escape Clause;     Content: ... |
|       (4) ID: 15177554;     Title:The Santa Clause;     Content: ... |
|       (5) ID: 9738451;     Title:The Santa Clause (film series) ;     Content: ... |
| } |
| **Judge Agent:** Sufficient Information. (Move to answer step) |
| **Final Query Pool:**{...} |
| **Answer Agent:** Jack Frost played by Martin Short. |
| Final Output is: Jack Frost played by Martin Short. |

Table 5: Case study illustrating a step-by-step perfect answer generating process of MAPS

| |
|---|
| **Case Study 2** |
| **Question:** How many games do team in the league where Barcelona won titles in 1948 and 1949 play? |
| **Ground truth:** 38 |
| **Retrieval Round 0:** |
| (Using question as query to retrieve) |
| **Query Pool:**{ |
|     Query 0: How many games do team in the league where Barcelona won titles in 1948 and 1949 play? |
|     Retrieved Chunks for Query 0: |
|     (1) ID: 5771248;   Title: UER Pineda de Mar;   Content: ... |
|     (2) ID: 12444826;   Title: 1949 Campeonato Profesional;   Content: ... |
|     (3) ID: 12905044;   Title: CB Viladecans;   Content: ... |
|     (4) ID: 6793510;   Title: CB Viladecans;   Content: ... |
|     (5) ID: 17705057;   Title: History of FC Barcelona;   Content: ... |
| } |
| **Judge Agent:** Insufficient Information. (Continue to the next retrieval round) |
| **Retrieval Round 1:** |
| **Retrieval Extension agent:** La Liga. (Using 'La Liga' as query to retrieve) |
| **Query Pool:**{ |
|     Query 0: How many games do team in the league where Barcelona won titles in 1948 and 1949 play? |
|     Retrieved Chunks for Query 0: ... |
|     Query 1: La Liga |
|     Retrieved Chunks for Query 1: |
|     (1) ID: 5401575;   Title: La Liga;   Content: ... |
|     (2) ID: 13260661;   Title: La Liga;   Content: ... |
|     (3) ID: 5404615;   Title: La Liga;   Content: ... |
|     (4) ID: 11762416;   Title: La Liga;   Content: ... |
|     (5) ID: 8344234;   Title: Liga de Futbol Profesional;   Content: ... |
| } |
| **Judge Agent:** Sufficient Information. (Move to answer step) |
| **Final Query Pool:**{...} |
| **Answer Agent:** 38 |
| **Final output is** 38. |

## D   LARGE LANGUAGE MODELS USAGE

We use large language models to refine grammar and enhance sentence clarity in our manuscript, as well as to aid in code debugging by explaining errors. This disclosure follows the ICLR 2026 policy on LLM usage.

