# OpenReview forum: "Multi-agent System with Individual Optimized Expertise for Retrieval Augmented Generation"
_ICLR.cc/2026/Conference — ICLR 2026 Conference Withdrawn Submission_

### Official Review · Reviewer_pVaw · 2025-10-30

**Soundness:** 3
**Presentation:** 3
**Contribution:** 3
**Rating:** 6
**Confidence:** 4

**Summary:**

This paper proposes MAPS, a novel multi-agent framework for Retrieval-Augmented Generation (RAG) that mitigates inadequate retrieval performance and hallucination issues. The system orchestrates three specialized agents: a Query Extension Agent for query refinement, a Judge Agent for quality assessment, and an Answer Agent for response generation. Comprehensive experiments across in-domain and out-of-domain benchmarks validate the effectiveness of the proposed approach.

**Strengths:**

1. The paper propose a multi-agent architecture where each agent is individually optimized for its specific subtask. Unlike existing multi-agent RAG methods that share a single backbone LLM, MAPS assigns task-adaptive LLMs to different agents, allowing for true specialization.

2. The paper introduces novel training strategies tailored to each agent, particularly the cross-agent collaboration where the answer agent's quality serves as supervision for training the query extension agent. This tight coupling between agents during training enhances coordination and leads to more effective retrieval and answer generation in the multi-agent system.

3. Comprehensive evaluation across both in-domain (HotpotQA, 2WikiMultiHopQA) and out-of-domain (Musique, Bamboogle) benchmarks demonstrate the effectiveness and generalization of the proposed method.

**Weaknesses:**

1. The paper completely omits inference time comparisons, which is critical given that the multi-agent architecture requires running three separate LLMs (Query Extension, Judge, Answer agents) plus potentially multiple retrieval rounds, inevitably increasing latency compared to single-model baselines. Without analyzing the performance-latency trade-off or providing metrics like average query response time and computational overhead, it's impossible to assess whether the reported performance gains justify the additional computational cost for practical deployment, especially in real-time applications where response time is crucial.

2. The paper lacks comprehensive discussion and comparison with recent multi-agent RAG systems. These works share similar motivations of using specialized agents/models for different RAG subtasks, including:
* Small Models, Big Insights: Leveraging Slim Proxy Models To Decide When and What to Retrieve for LLMs
* C-3PO: Compact Plug-and-Play Proxy Optimization to Achieve Human-like Retrieval-Augmented Generation
* MA-RAG: Multi-Agent Retrieval-Augmented Generation via Collaborative Chain-of-Thought Reasoning

A more thorough comparison would better position MAPS's contributions and clarify its advantages over these approaches, particularly in terms of training methodology and performance gains.

**Questions:**

1. What is the average latency per query compared to baseline methods?
2. What percentage of queries require multiple retrieval rounds, and what's the distribution of retrieval iterations?

---

### Official Review · Reviewer_6PzV · 2025-11-01

**Soundness:** 2
**Presentation:** 3
**Contribution:** 2
**Rating:** 2
**Confidence:** 4

**Summary:**

This paper introduces a new approach to optimizing a multi-agent system with individually optimized expertise (MAPS) for retrieval-augmented generation (RAG). Specifically, the authors propose equipping three specialized agents with individual optimization corpora and learning strategies to enhance retrieval quality (via the query extension agent and judge agent) and mitigate generation hallucination (via the answer agent). The three agents—the answer agent, query extension agent, and judge agent—are optimized sequentially and then collaborate to produce the final output within a standard RAG pipeline. The experimental results show improvements over the selected baselines, and the ablation studies further verify the effectiveness of each optimized agent. Overall, the paper is well-written, and the proposed method is clearly described.

**Strengths:**

The paper is well-structured and clearly written, making it easy to follow the proposed methodology and experimental setup. The proposed MAPS framework is logically designed, with each agent having a specific role and optimization strategy. The design of adopting different optimization algorithms for different agents is interesting and could potentially lead to better performance in RAG tasks. The experimental results show improvements over several baselines, and the ablation studies provide insights into the contributions of each agent.

**Weaknesses:**

1. The novelty of individual agent optimization is rather limited. Multi-agent systems for RAG have been explored in prior work, and the idea of optimizing agents individually is not entirely new. Recent studies have increasingly focused on optimizing multiple agents within the RAG pipeline, and individual agent optimization has also been investigated as a preliminary stage before joint optimization. Therefore, the authors should more clearly articulate which specific aspects of their approach are novel compared to existing multi-agent RAG methods.
2. The motivation and theoretical analysis are insufficient. While the paper provides empirical results, it lacks a thorough theoretical discussion or justification for why and how individual agent optimization can lead to better performance in RAG tasks. Moreover, the authors adopt different optimization algorithms for each agent (e.g., GRPO for the answer and query extension agents, and SFT for the judge agent), but there is no clear rationale provided for these specific choices. The reward design is also insufficiently justified.
3. The experimental comparisons are insufficient. The experimental section lacks comparisons with several recent and relevant multi-agent RAG methods. Including these comparisons would provide a clearer and more convincing evaluation of the proposed method’s effectiveness relative to the state of the art.

**Questions:**

see weaknesses.

---

### Official Review · Reviewer_hMJE · 2025-11-02

**Soundness:** 2
**Presentation:** 2
**Contribution:** 3
**Rating:** 2
**Confidence:** 4

**Summary:**

This paper introduces MAPS, a RAG framework that decomposes the standard single-agent RAG pipeline into three specialized agents: a query extension agent, a context sufficiency judge, and an answer generator. Specifically, the answer agent is trained with a two-stage GRPO to learn to minimize internal hallucinations of the model and omit external noisy context separately. The query-extension agent is trained with GRPO together with a well-trained answer agent using a delta-F1 reward to generate effective queries. The judge agent is trained with SFT on the offline dataset collected using Debate RAG. The workflow involves iterative retrieval guided by the query agent, a binary decision by the judge on whether enough context has been collected, and final answer generation conditioned on the aggregated retrieved evidence. Across four QA benchmarks, MAPS consistently exceeds strong single‑ and multi‑agent RAG baselines—including recent R1‑style reasoning systems—and ablations indicate that each agent contributes, with the answer agent most critical and the judge improving out‑of‑domain robustness.

**Strengths:**

1. The idea of using multi-agent to solve subtasks for RAG and optimize them separately is simple yet effective
2. Thorough ablation studies, including agent removal, mixed-vs-single reward, and retrieval round analyses.

**Weaknesses:**

1. **Non-anonymous links**: At line 368, the authors include a Hugging Face link (https://huggingface.co/Angus998/MAPS) that may violate the double-blind review policy.
2. **Clarity of dataset construction and GRPO training**: Section 3 is written at a high level and focuses on reward design, but omits crucial details about dataset construction and the RL training objective. For example, in Sec. 3.2, the answer-agent's training set is said to be built with a retrieval-only debate RAG but missing key procedural details - e.g., what is exactly the debate procedure; how snapshots/rounds are selected or weighted; or statistics of the final training set (e.g., numbers of queries/chunks/rounds, and whether those snapshots are “sufficient” per the debate‑stage judge). In Sec. 3.3, the construction of the query‑refinement dataset is unclear - e.g., does it contain only a sampled subset of HotpotQA and 2WikiMultiHopQA questions, or further preprocessing required to generate refined queries? It would be beneficial for the paper’s clarity and reproducibility if the authors explained these points in detail.
3. **Lack explanation of reward design**: For the answer agent, why use Precision during stage 1 and switch to F1 during stage 2? For the query extension agent, the step-wise $\Delta(F1)$ reward in Eq. 7 may have a shrinking upper bound as $n$ increases; if so, how does this affect GRPO training?
4. **Judge-labeling**: It's unclear why the authors choose to label the judge using the current unanimity rule, i.e., all‑correct $\rightarrow$ “sufficient”, all‑incorrect $\rightarrow$ “insufficient”. What if the 16 sampled answers are partially correct? Is this one of the reasons that leads to overfitting of the judge agent as shown in Table 2?
5. **Backbone diversity**: All agents are trained and evaluated with `Llama-3.1-8B-Instruct` as the backbone, which may introduce bias and raise concerns about the generalizability of the finding.
6. **Training-inference mismatch**: The training corpora for the answer and judge agents are produced by a retrieval‑only debate RAG pipeline, and debate snapshots become training samples; at inference, however, MAPS runs a simple sequential loop (query‑extension $\rightarrow$ judge $\rightarrow$ answer) with no debate. Moreover, Algorithm 1 shows each agent is trained separately: the answer agent $A_g$ on debate data (T1), the query agent $A_q$ on a prebuilt single‑retrieval dataset (T2) with rewards computed by a well-trained $A_g$, and the judge $A_j$ via SFT from labels generated by that same $A_g$ (T3). The paper does not analyze how this distributional shift (debate vs. sequential) or the off‑policy nature of $A_q$ and $A_j$ training affects stability or generalization.

**Questions:**

Please see the weakness section. In addition, I have the following questions:

1. Following W.2, for the training of answer agent, when synthesize the dataset, if a lot of snapshots at early rounds are selected (with insufficient context), will this eventually results in training the answer agent to incentivize its own reasoning and knowledge without paying that much attention on the retrieve context which potentially results in the dominant performance of the answer agent.
2. **Reward training dynamic**: Figure 3 seems to show limited improvement in the training reward. Can the authors clarify the learning dynamics?
3. **Reward hyper-params**:  What values were used for $\alpha$ and $l$? If possible, please include a sensitivity analysis of these hyperparameters.
4. Following W.6, is it feasible to train the three agents on‑policy in an end‑to‑end manner? E.g., $A_q$ generates actual queries during training, with $A_j$ gating rounds, and optimize all the agents together on the resulting traces using the overall EM/F1 of $A_g$'s prediction (maybe also with penalties for extra rounds/length) as the reward?
5. **Discrepancy in experiment setting**: This is a minor thing. Sec. 3.2 states that the retrieve-only debate RAG is built with `Llama-3-8B-Instruct` whereas Sec. 4.1 lists `Llama-3.1-8B-Instruct` as the backbone. Please clarify this mismatch.

---

> ### Author Response · Authors · 2025-11-20
>
> We would like to clarify that the link-related Hugging Face account and repository were created specifically for this submission. The username does not reflect the real names or affiliations of the authors, and no personal information is present on either the account or model page. Its sole purpose is to facilitate model availability for reproducibility and further research.
>
> We sincerely thank you for your careful and detailed review. We will carefully revise and improve the work based on your suggestions.

---

### Note · Authors · 2025-11-20

I have read and agree with the venue's withdrawal policy on behalf of myself and my co-authors.